# Early Patient-Triggered Pressure Support Breathing in Mechanically Ventilated Patients with COVID-19 May Be Associated with Lower Rates of Acute Kidney Injury

**DOI:** 10.3390/jcm12051859

**Published:** 2023-02-26

**Authors:** Mark E. Seubert, Marco Goeijenbier

**Affiliations:** 1Intensive Care, The LangeLand Hospital, 2725 NA Zoetermeer, The Netherlands; 2Intensive Care, Spaarne Gasthuis, 2035 RC Haarlem, The Netherlands; 3Intensive Care, Erasmus MC, 3015 GD Rotterdam, The Netherlands

**Keywords:** COVID-19, mechanical ventilation, acute kidney injury (AKI), acute respiratory distress syndrome (ARDS), ventilator-induced kidney injury (VIKI)

## Abstract

Background: Acute respiratory distress syndrome (ARDS) in COVID-19 patients often necessitates mechanical ventilation. Although much has been written regarding intensive care admission and treatment for COVID-19, evidence on specific ventilation strategies for ARDS is limited. Support mode during invasive mechanical ventilation offers potential benefits such as conserving diaphragmatic motility, sidestepping the negative consequences of the longer usage of neuromuscular blockers, and limiting the occurrence of ventilator-induced lung injury (VILI). Methods: In this retrospective cohort study of mechanically ventilated and confirmed non-hyperdynamic SARS-CoV-2 patients, we studied the relation between the occurrence of kidney injury and the decreased ratio of support to controlled ventilation. Results: Total AKI incidence in this cohort was low (5/41). In total, 16 of 41 patients underwent patient-triggered pressure support breathing at least 80% of the time. In this group we observed a lower percentage of AKI (0/16 vs. 5/25), determined as a creatinine level above 177 µmol/L in the first 200 h. There was a negative correlation between time spent on support ventilation and peak creatinine levels (r = −0.35 (−0.6–0.1)). The group predominantly on control ventilation showed significantly higher disease severity scores. Conclusions: Early patient-triggered ventilation in patients with COVID-19 may be associated with lower rates of acute kidney injury.

## 1. Introduction

Critical care practitioners debate when or whether to allow patient-triggered ventilation in acute respiratory distress syndrome (ARDS) patients. A recent Cochrane intervention review by Hohmann et al. pointed out that there is a lack of data on the benefits and harms of early supported ventilation in invasively ventilated persons with COVID-19 [1].

Invasive mechanical ventilation, in any form, is associated with a threefold increased risk of developing acute kidney injury (AKI), and no direct association is seen between various tidal volume settings or level of positive end-expiratory pressure (PEEP) [2]. As early as 1947, Drury et al. published the first study of the effects of mechanical ventilation on renal function. Studying the effect of varying levels of continuous positive airway pressure on urea clearance in healthy subjects suggested the kidney–lung connection [3]. Although not completely understood and even harder to predict, the systemic effects of mechanical ventilation suggest potential mechanisms of ventilator-induced kidney injury (VIKI) and suggest that these mechanisms might be predicted using biomarkers [4]. Ideas on why ventilator-induced kidney injury can occur include the fact that mechanical ventilation may induce rapid hemodynamic changes, neurohumoral mediated alterations in intrarenal blood flow, and systemic inflammatory cytokines released systemically due to ventilator-induced lung injury [4]. Overall, invasive mechanical ventilation is a proven risk factor for the occurrence of early AKI [5]. 

Especially in prolonged ICU stays, as seen in severe COVID-19, it is important to differentiate between early AKI that is directly related to the primary pathology and treatment (i.e., within the first 14 days of ICU admittance) versus late AKI, which is predominantly related to other causes such as superinfections and drug toxicity [6,7]. A recent study suggested a relation between the occurrence of AKI and the level of PEEP used. Of specific interest is that nearly all AKI patients in this study were treated with neuromuscular blockade (NMB) (97 vs. 80%, respectively), and thus there were no patient-triggered breathing efforts [8]. We hypothesized that the high incidence of AKI in COVID-19 patients is related to a high percentage of time spent on controlled ventilation in the first 200 h after starting invasive respiratory support. To investigate this, we analyzed the patient data from a single-center intensive care unit in Zoetermeer, the Netherlands.

## 2. Materials and Methods

We evaluated a retrospective cohort of COVID-19 patients admitted to the intensive care unit between March 2020 and March 2022 in the LangeLand Hospital in Zoetermeer, the Netherlands. The aim of this study was to assess the relation between the incidence of AKI and the ratio of pressure support to control (PS/PC + PS ratio) in invasive mechanical ventilation. The KDIGO (Kidney Disease: Improving Global Outcomes) guidelines are used to define AKI.

Data from the first 200 h of mechanical ventilation were used, and the ratio for each patient was determined by dividing the hours spent on pressure support by the total hours spent on mechanical ventilation (pressure support + pressure control). Graph Pad Prism 9.0 was used for statistical analyses. The ethics committee of the LangeLand Hospital approved the study.

## 3. Results

In total, 130 patients with COVID-19 were admitted to intensive care during the study period. Of these, 46 patients needed invasive mechanical ventilation, of which 41 were not transferred to another hospital within the investigated timeframe (i.e., were transferred to another hospital within 14 days of intensive care admittance or were admitted from another intensive care and already ventilated for longer than 3 days). Prone positioning was initiated when the pao2/fio2-ratio < 150 (while maintaining pressure support if not previously on pressure control setting). Interleukine-6 inhibitors and dexamethasone were administered as soon as evidence for their beneficial effects was established. Patient characteristics of interest are summarized in Table 1. In total, 16 of 41 patients underwent patient-triggered pressure support breathing at least 80% of the time during invasive mechanical ventilation (meaning a PS to PC ratio of 0.8). Briefly, the significant differences between a PS/PC + PS ratio above 80% versus below 80% were the Apache IV score (50 ± 14.5 vs. 66 (±17.9), SAPS II score (25 ± 0 vs. 31 (±9), and the number of patients with a creatinine level above 177 µmol/L in the first 200 h (0/16 vs. 5/25). Based on current guidelines, three of the forty-one patients required renal replacement therapy. However, this was deemed futile and not started in two cases, based on comorbidity and pre-disease performance.

The group with a higher percentage of controlled mechanical ventilation had a higher incidence of AKI, defined as a creatinine level above 177 µmol/L (5/25 vs. 0/16) within the first 200 h of mechanical ventilation. However, this group also showed higher disease severity based on the increased APACHE IV and SAPS scores. When disease severity scores were corrected for AKI, the difference decreases but remains significant. In the cohort as a whole, the PS/PC + PS ratio was negatively correlated with the peak level of creatinine (r = −0.35 (−0.6–0.1) *p* 0.03) (see Figure 1), meaning that time spent on supported ventilation was negatively correlated with peak creatinine levels.

## 4. Discussion

In this retrospective cohort study, a significant negative correlation was observed between peak creatinine levels and the predominance of controlled ventilation. Furthermore, we observed a remarkably low level of AKI compared to other intensive care patients. The occurrence of AKI in the early phase of COVID-19 disease can be the result of many factors. Overall, five patients (12.2%) showed an increase in creatinine above 177 µmol/L within 14 days after intensive care admittance, and only three (7.3%) potentially qualified for renal replacement therapy. However, in two cases RRT was not initiated due to treatment restrictions based on comorbidity and pre-disease performance. If we look at the complete cohort of mechanically ventilated COVID-19 patients, we find a much lower incidence of AKI compared to past data for this relation, i.e., ranging from 20% to as high as 90% [2,9,10,11,12].

In this cohort, as in most intensive care settings treating COVID-19 patients, several patients were ventilated with high levels of PEEP, up to 22 cmH_2_O, and required prone positioning for longer periods of time. We found it possible to also have these patients ventilated on a patient-triggered assisted mode if sedation and analgesics were dosed accordingly. This is done through continuously monitoring end tidal CO_2_ and performing blood-gas analysis regularly, in addition to assuring patient comfort. Clearly, hyperventilation would be an initial sign of discomfort, and hypoventilation often would result in decreasing the dosage of intravenous opiates.

A retrospective cohort study on an intensive care unit comes with many potential pitfalls and possible alternative explanations. First, a potential reason for the low AKI incidence is that patients in our cohort were less ill compared to the published cohorts. Based on APACHE IV scores (Table 1) and the fact that there was no selection of patients referred to the hospital during the pandemic, this is considered less likely. Another explanation could be that more severely ill patients were selected for transfer to other hospitals, diluting the case severity mix. However, the APACHE scores from patients in our cohort did not differ from national data from a national cohort study [13]. One factor that could influence the potential explanation for the lower AKI incidence could be the high percentage of patient-triggered breathing allowed. We hypothesize that increasing the time spent on support ventilation, more specifically patient-triggered assisted ventilation, could decrease the incidence of ventilator-induced kidney injury. This idea is supported by the significant negative relation between the time spent on assisted ventilation and peak creatinine levels. However, we are aware that our limited data must be interpreted with caution. The biggest shortcoming of this retrospective study is the difference in disease severity within this cohort, of which only a small part can be explained by the change in renal function. One could argue that comparison within this small cohort is not feasible. Alternative explanations such as comorbidities, fluid overload, vasopressor use, and nephrotoxic drugs all contribute to the incidence of AKI on the ICU. Taking this in mind, statistical results from the comparison of two groups in this small cohort might not be very useful and could be prone to bias. However, overall, the low AKI incidence and the significant relation between renal function and time spent on pressure support the idea of a renal–pulmonary interaction.

The association between controlled mechanical ventilation and AKI is of great interest and has been studied before [3,4]. The COVID-19 pandemic gives us more opportunities to compare choices made in mechanical ventilation and their effects on renal function. AKI in COVID-19 shows a relatively high incidence, as much as double that in other viral respiratory diseases such as influenza [14,15]. The suggested pathophysiology of AKI in COVID currently relies on nonspecific mechanisms (hypovolemia, nephrotoxic drugs, high PEEP, right heart failure), direct viral injury, imbalanced renin–angiotensin–aldosterone system (RAAS) activation, elevation of proinflammatory cytokines elicited by viral infection, and a profound procoagulant state [14,16]. In addition to the need for understanding the concept of renal interstitial edema when treating patients in general, limiting intravenous fluids significantly and prioritizing pressure-support modality over pressure or volume control settings could play a pivotal role. COVID-19 patients predominantly present with single-organ failure, and compared to patients with sepsis, exhibit only a limited increase in catabolic state, measured through the necessary respiratory minute volume (MV) when sufficiently sedated and anaesthetized. The MV rarely exceeds 12 L and mostly remains under 10 L without inducing hypercapnia. Therefore, in pathology limited to COVID-19, or perhaps even within other patient groups in a non-hyperdynamic state, the work of breathing is not excessive when ventilated in a pressure support manner. Therefore, it is necessary to titrate sedation and analgesia to this aim. Another contributing explanation could be the effects of neuromuscular blocking agents when used for longer periods of time. Marchiset et al. reported a significant increased risk of the development of AKI during intensive care stay associated with the use of these agents. Naturally, with the use of neuromuscular blocking agents patient-triggered breaths will be nearly absent [17].

Finally, although based on a relatively small number of patients, and even considering the potential alternative explanations for some of the AKI cases, compared with the existing literature our observations support evaluation through future studies to explore the beneficial relation between predominantly supportive ventilation and a decreased incidence in kidney injury. We feel this may be an important contributing factor explaining the increased incidence of AKI in mechanically ventilated patients, especially patients with COVID-19, and possibly even for other patients, after the suggested relation has been confirmed.

## Figures and Tables

**Figure 1 jcm-12-01859-f001:**
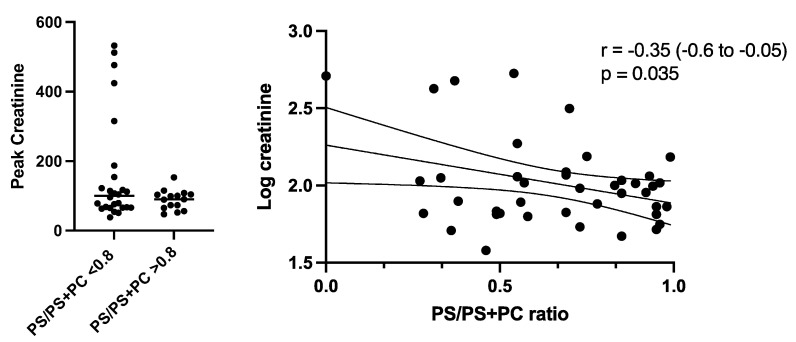
First panel shows the peak creatinine levels in µmol/L during the ICU stay in patients with a ratio (PS/PC + PS) above 80% and below 80%. A trend is seen but is not statistically significant (*p* = 0.072). In the second panel a significant negative correlation (r = −0.35) is seen between peak creatinine levels and the predominance of controlled ventilation.

**Table 1 jcm-12-01859-t001:** Baseline characteristics of 41 patients undergoing invasive mechanical ventilation due to COVID-19 ARDS. Numbers represent the mean ± standard deviation.

	PS-PC Ratio > 0.8	PS-PC Ratio < 0.8	*p* Value
n	16	25	
Age	60 (47–73)	62 (54–70)	*p* = 0.79 NS
Days in ICU	20 ± 18	14 (±8)	*p* = 0.51 NS
90-day mortality	1/16	5/25	*p* = 0.2 NS
APACHE IV score	50 (±14.5)	66 (±17.9)	*p* = 0.008 *
SAPS II	25 (±9)	31 (±9)	*p* = 0.02 *
APACHE IV, no AKI	47 (±14.5)	62 (±16)	*p* = 0.009 *
SAPS II, no AKI	24 (±8)	31 (±9)	*p* = 0.01 *
Peak creatinin in µmol/L	88 (±28)	160 (±150)	*p* = 0.2
Creatinin > 177 µmol/L in first 200 h	0/16	5/25	*p* = 0.05 *
PC hours first 200 h	13 (±10)	80 (±35)	*p* = 0.0001 *
Average PEEP ^§^	14.31	15.25	*p* = 0.24

^§^ Set PEEP (cmH2O) with unadjusted setting for at least 4 h to exclude measurements from recruitment procedures, etc. * *p* ≤ 0.05.

## Data Availability

The data presented in this study are available on request from the corresponding author.

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
