# Peer review of "Early Patient-Triggered Pressure Support Breathing in Mechanically Ventilated Patients with COVID-19 May Be Associated with Lower Rates of Acute Kidney Injury"

_jcm, 2023, doi:10.3390/jcm12051859_

Round 1

Reviewer 1 Report

The work provided by Seubert et al deals with the importance of the beneficial effects of spontaneous breathing on renal function in COVID-19 patients. The authors compare two populations with invasive mechanical ventilation depending on the pressure support and pressure control ratio. The authors furthermore describe the beneficial effect of spontaneous breathing on renal function. Although I am a supporter of spontaneous breathing in the ICU patient, the conclusions of the authors must be taken with caution.

1.     The two populations are, in my opinion, too different. This is supported by the APACHE and SAPS II scores. Although the authors have corrected for AKI, these differences still stand.

2.     no information on the state of patient hemodynamics is provided. It is essential to know the history of vasopressive support and volume status.

3.     What was the cognitive status of the patients, awake vs sedated?

Some minor revisions should be addressed

In lines 73-74 the authors write “creatinine level above 177 (7/19 versus 0/15; p = 0.035) within the first 200 hours of mechanical ventilation” however, table 1 states 0/16 and 7/26. How is this to be interpreted?

Some minor grammar and styling should be addressed

In line 49: Lange Lang Ziekenhuis whereas in line 52: LangeLand Ziekenhuis

Author Response

Response to Reviewer 1:

First, we would like to thank Reviewer 1 for taking the time to thoroughly review our manuscript. His or her suggestions significantly improved the quality of our short communication and made us realize we needed to rephrase our message based on our available data.

We absolutely agree with reviewer 1 that dividing this relative small retrospective cohort in two groups might not make sense. This led to two groups with significant differences in disease severity. However, we believe that in our data suggests an important message with large implications if confirmed. I.e., through starting  a prospective study on supported versus controlled mechanical ventilation. We decided to pursue publishing this data, although far from perfect, because of the very low incidence of AKI in our cohort compared to national and international data. Based on our work we cannot draw firm conclusions, but it does deliver food for thought. Furthermore, we believe the correlation between creatinine levels and the support to controlled ratio, in line with important earlier work by others, justifies the design of such a study resulting in a higher level of evidence. Taken this in mind we decided to publish a short commentary adding data for the need for higher level of evidence between the relation of kidney function and controlled mechanical ventilation.

Taking the suggestions by Reviewer 1 in mind we rewrote parts of the manuscript. With more attention to the potential pitfalls in our design. Focussing more on comparison with (inter)national data and the correlation within the whole cohort. Rather than comparing two groups.

Minor suggestions:
In lines 73-74 the authors write “creatinine level above 177 (7/19 versus 0/15; p = 0.035) within the first 200 hours of mechanical ventilation” however, table 1 states 0/16 and 7/26. How is this to be interpreted?

Response to the reviewer: we like to thank reviewer 1 for taking the time for a detailed review of our work. Indeed, we made a mistake by taking the group as a whole. We adjusted the numbers and re checked with the database.

In line 49: Lange Lang Ziekenhuis whereas in line 52: LangeLand Ziekenhuis

Response: changed accordingly

Reviewer 2 Report

1. The negative impact of IMV on renal function is known since publication of Drury et al. in 1947. Now the authors of the manuscript present potentially interesting finding of assumed beneficial impact of higher ratio of triggered breaths to controlled breaths on the prevalence of AKI in mechanically ventilated patients. Unfortunately, the data presented in the manuscript can hardly support this hypothesis.

2. Abbreviations should be defined

3.

Materials and Methods

What definition of AKI was used in the manuscript?

What stage of AKI was used?

How pressure support to control ratio during mechanical ventilation was determined?

Results

Line 51-52: Place and date of the study should be in the „Materials and Methods” section.

Line 65-67: This sentence is hardly comprehensible and it should be reworded.

Line 70-71: I can’t find what does the note apply to, and what does it try to explain?

Line 72-74: The numbers are different from those given in the table. Information on units for creatinine level is missing.

Table 1:

Co-morbidities are very important predictors of AKI and there should be information on number of patients with e.g. diabetes, hypertension and septic shock in both groups. How many patients were on vasopressors in both groups?

creatinine – units?

What is: „AP mortality chance”?

Discussion

Line 93-94: I guess the authors meant: „other intensive care units”.

Other potential reasons for the difference of AKI between the groups should be discussed, for example co-morbidities, fluid balance, use of nephrotoxic drugs etc.

4.

Conclusion:

The two groups were hardly comparable due to the significant difference in APACHE and SAPS II score. Moreover 3 out of 7 patients with high creatinine level in the „group B” had treatment limitations which accounts for 42,8%. This could explain the higher number of patients with AKI in the PS-PC ratio <0.8 group. The hypothesis about the beneficial impact of higher ratio of triggered breaths to controlled breaths on the prevalence of AKI in mechanically ventilated patients needs more arguments.

The conclusion of the presented data is included in the title of the manuscript, and in my opinion the title: „Early spontaneous breathing in mechanically ventilated patients with COVID-19 and lower rates of acute kidney injury” is more justified.

Author Response

First, we would like to thank reviewer 2 for his or her valuable suggestions and the time taken for extensively reviewing our communication. Starting with historic knowledge and adding the first rightful citation on the negative impact of IMV on renal function.

The major revisions suggested by reviewer 2 all seem valid and increase the quality of our work. Reading the concerns of reviewer 2 made us realize that in the writing process we have over interpreted our data and the comparison of two small groups in a small, retrospective cohort does not help to get our message through make a lot of sense. We have rewritten parts of the manuscript to better formulate rephrase our message. Putting more attention to the limitations and pitfalls and adapting our conclusions.

What we noticed in our retrospective cohort is the absolute lower incidence of AKI. Compared to other Dutch intensive cares and compared to international literature. We then sought to explain this low incidence and came to the hypothesis of our high incidence of supported ventilation, again compared to literature. Although, we absolutely agree with reviewer 2 that no hard conclusions can be drawn from our cohort. Especially, taking all his or her potential other explanations in mind, we do believe the data, as it is, is worth publishing as a communication. This is because, in the whole cohort the correlation between the PS to PC ratio with creatinine levels adds to the available, but limited, evidence on the impact of IMV on renal function. But even more important, it absolutely encourages the start of a prospective study on the role of IMV on renal function.

Minor suggestions:
The negative impact of IMV on renal function is known since publication of Drury et al. in 1947. 

Answer: added the manuscript of Drury. Thanks!

Abbreviations should be defined

Answer: changed accordingly 

Materials and Methods

What definition of AKI was used in the manuscript?

Answer: We agree with reviewer 2 this should be explained in the manuscript. However, we wanted to be concise as possible, publishing this data as a communication. The extra word count made available by the journal gave us the opportunity to increase the quality of our methods section.
KDIGO, added to the part of the methods

What stage of AKI was used?
Acute stage (not chronic kidney disease)

How pressure support to control ratio during mechanical ventilation was determined?
Answer: we retrieved the hours spent on either pressure support or pressure control within the first 200 hours of intensive care admission from our PDMS (patient data management system). The ratio was calculated by PS/PS+PC. We explained this in more detail in the revised version.

Results

Line 51-52: Place and date of the study should be in the „Materials and Methods” section.

Answer: changed accordinly

Line 65-67: This sentence is hardly comprehensible and it should be reworded.

Answer: absolutely agree with reviewer 2. Rewrote this part of result section.

Line 70-71: I can’t find what does the note apply to, and what does it try to explain?

Answer: rewrote this sentence

Line 72-74: The numbers are different from those given in the table. Information on units for creatinine level is missing.

What is AP mortality chance:
Answer: locally used, deleted from table, did not add to the manuscript other than SAPS and APACHE

Comorbidities:

No significant difference between the groups is present on pre ICU renal disease. We dont have data on hypertension nor DM type 2. We have listed this as potential shortcomings in the discussion.

Conclusion:

Answer to the reviewer:

We rewrote our discussions and conclusions and changed our title based on your suggestions. We would like to thank you for your time.

Round 2

Reviewer 1 Report

Thank you for the revision of the manuscript, it has certainly improved the overall quality of the work. 

I do however, have some minor issues: as in my previous review, the data provided in table 1 and lines 100-101 do not match, either the correction was not done or you have submitted the wrong version. 

Furthermore, several minor grammar issues have to be addressed as well as uniformity in the references (Ref.3 does not match the rest).

Author Response

Dear reviewer 1,

We would like to thank you for your positive comments.

We have addressed the minor comments, typo's, refs and the data matches table 1 now (indeed wrong upload, apologies)

Reviewer 2 Report

Line 88 and others: Please indicate which iteration of APACHE score is used? Is it APACHE II or APACHE IV?

Line 89: it should be: „177 µmol/l”

Line 89: 7/15 vs. 0/15 is not compatible with the numbers given in the table: 7/26 and 0/16

Table 1: The cohort consists of 41 patients divided in two groups: 16 and 26 patients, but 16+26 = 42 and not 41

Instead: „Kreatinin>177 in first 200 hours”, should be: „Creatinine>177 µmol/l in first 200 hours”

Line 101: again 7/15 versus 0/15 is not compatible with the numbers given in the table: 7/26 and 0/16

Line 105: instead „r = 035”, it should be: r = 0.35

Line 121: The number of patients with creatinine level above 177 µmol/l mentioned in the text is five, compared to seven in the table

Line 151: instead „ventilator induces kidney injury”, should be „ventilator induced kidney injury”

Author Response

Dear reviewer 2,

many thanks for again taking the time to review our manuscript.

We have changed all suggestions accordingly in the revised version of our manuscript. 

Thanks again